# Enhancing Low-resource Fine-grained Named Entity Recognition by Leveraging Coarse-grained Datasets

**Su Ah Lee**[1†]**, Seokjin Oh**[1†]**,** and **Woohwan Jung**[1,2]

[1]Department of Applied Artificial Intelligence, Hanyang University
[2]Ramply Inc.
{sue991, seokjinoh, whjung}@hanyang.ac.kr

## Abstract

Named Entity Recognition (NER) frequently suffers from the problem of insufficient labeled data, particularly in fine-grained NER scenarios. Although $K$-shot learning techniques can be applied, their performance tends to saturate when the number of annotations exceeds several tens of labels. To overcome this problem, we utilize existing coarse-grained datasets that offer a large number of annotations. A straightforward approach to address this problem is pre-finetuning, which employs coarse-grained data for representation learning. However, it cannot directly utilize the relationships between fine-grained and coarse-grained entities, although a fine-grained entity type is likely to be a subcategory of a coarse-grained entity type. We propose a fine-grained NER model with a Fine-to-Coarse(F2C) mapping matrix to leverage the hierarchical structure explicitly. In addition, we present an inconsistency filtering method to eliminate coarse-grained entities that are inconsistent with fine-grained entity types to avoid performance degradation. Our experimental results show that our method outperforms both $K$-shot learning and supervised learning methods when dealing with a small number of fine-grained annotations. Code is available at https://github.com/sue991/CoFiNER.

## 1 Introduction

Named Entity Recognition (NER) is a fundamental task in locating and categorizing named entities in unstructured texts. Most research on NER has been conducted on coarse-grained datasets, including CoNLL'03 (Tjong Kim Sang, 2002), ACE04 (Mitchell et al., 2005), ACE05 (Walker et al., 2006), and OntoNotes (Weischedel et al., 2013), each of which has less than 18 categories. As the applications of NLP broaden across diverse fields, there is increasing demand for fine-grained NER that can provide more precise and

detailed information extraction. Nonetheless, detailed labeling required for large datasets in the context of fine-grained NER presents several significant challenges. It is more cost-intensive and time-consuming than coarse-grained NER. In addition, it requires a high degree of expertise because domain-specific NER tasks such as financial NER (Loukas et al., 2022) and biomedical NER (Sung et al., 2022) require fine-grained labels. Thus, fine-grained NER tasks typically suffer from the data scarcity problem. Few-shot NER approaches (Ding et al., 2021) can be applied to conduct fine-grained NER with scarce fine-grained data. However, these methods do not exploit existing coarse-grained datasets that can be leveraged to improve fine-grained NER because fine-grained entities are usually subtypes of coarse-grained entities. For instance, if a model knows what `Organization` is, it could be easier for it to understand the concept of `Government` or `Company`. Furthermore, these methods often experience early performance saturation, necessitating the training of a new supervised learning model if the annotation extends beyond several tens of labels.

A pre-finetuning strategy (Aghajanyan et al., 2021; Ma et al., 2022a) was proposed to overcome the aforementioned problem. This strategy first learns the feature representations using a coarse-grained dataset before training the fine-grained model on a fine-grained dataset. In this method, coarse-grained data are solely utilized for representation learning; thus, it still does not explicitly utilize the relationships between the coarse- and fine-grained entities.

Our intuition for fully leveraging coarse-grained datasets comes mainly from the hierarchy between coarse- and fine-grained entity types. Because a coarse-grained entity type typically comprises multiple fine-grained entity types, we can enhance low-resource fine-grained NER with abundant coarse-grained data. To jointly utilize both datasets, we

---

[†]Major in Bio Artificial Intelligence

devise a method to build a mapping matrix called F2C (short for 'Fine-to-Coarse'), which connects fine-grained entity types to their corresponding coarse-grained entity types and propose a novel approach to train a fine-grained NER model with both datasets.

Some coarse-grained entities improperly match a fine-grained entity type because datasets may be created by different annotators for different purposes. These mismatched entities can reduce the performance of the model during training. To mitigate this problem, coarse-grained entities that can degrade the performance of fine-grained NER must be eliminated. Therefore, we introduce a filtering method called 'Inconsistency Filtering'. This approach is designed to identify and exclude any inconsistent coarse-to-fine entity mappings, ensuring a higher-quality training process, and ultimately, better model performance. The main contributions of our study are as follows:

- We propose an F2C mapping matrix to directly leverage the intimate relation between coarse- and fine-grained types.

- We present an inconsistency filtering method to screen out coarse-grained data that are inconsistent with the fine-grained types.

- The empirical results show that our method achieves state-of-the-art performance by utilizing the proposed F2C mapping matrix and the inconsistency filtering method.

## 2 Related Work

**Fine-grained NER**. NER is a key task in information extraction and has been extensively studied. Traditional NER datasets (Tjong Kim Sang, 2002; Ritter et al., 2011; Weischedel et al., 2013; Derczynski et al., 2017) address coarse-grained entity types in the general domain. Recently, domain-specific NER has been investigated in various fields (Hofer et al., 2018; Loukas et al., 2022; Sung et al., 2022). The domain-specific NER tasks typically employ fine-grained entity types. In addition, Ding et al. (2021) proposed a general domain fine-grained NER.

$N$-**way** $K$-**shot learning for NER**. Since labeling domain-specific data is an expensive process, few-shot NER has gained attention. Most few-shot NER studies are conducted using an $N$-way $K$-shot episode learning (Das et al., 2021; Huang

et al., 2022; Ma et al., 2022b; Wang et al., 2022). The objective of this approach is to train a model that can correctly classify new examples into one of $N$ classes, using only $K$ examples per class. To achieve this generalization performance, a large number of episodes need to be generated. Furthermore, to sample these episodes, the training data must contain a significantly larger number of classes than $N$. In contrast, in our problem context, the number of fine-grained classes we aim to identify is substantially larger than the number of coarse-grained classes in the existing dataset. Consequently, the episode-based few-shot learning approaches are unsuitable for addressing this problem.

**Leveraging auxiliary data for information extraction**. Several studies have employed auxiliary data to overcome the data scarcity problem. Jiang et al. (2021); Oh et al. (2023) propose NER models trained with small, strongly labeled, and large weakly labeled data. Jung and Shim (2020) used strong and weak labels for relation extraction. However, in the previous studies, the main data and auxiliary data shared the same set of classes, which is not the case for fine-grained NER with coarse-grained labels. While the approaches of Aghajanyan et al. (2021) and Ma et al. (2022a) can be applied to our problem setting, they utilize auxiliary data for representation learning rather than explicitly utilizing the relationship between the two types of data.

## 3 Proposed Method

In this section, we introduce the notations and define the problem of fine-grained NER using coarse-grained data. Then, we introduce the proposed CoFiNER model, including the creation of the F2C mapping matrix.

### 3.1 Problem definition

Given a sequence of $n$ tokens $\mathbf{X} = \{x_1, x_2, ..., x_n\}$, the NER task involves assigning type $y_i \in E$ to each token $x_i$ where $E$ is a predefined set of entity types. In our problem setting, we used a fine-grained dataset $\mathcal{D}^{\mathcal{F}} = \{(\mathbf{X}_1^{\mathcal{F}}, \mathbf{Y}_1^{\mathcal{F}}), ..., (\mathbf{X}_{|\mathcal{D}^{\mathcal{F}}|}^{\mathcal{F}}, \mathbf{X}_{|\mathcal{D}^{\mathcal{F}}|}^{\mathcal{F}})\}$ with a predefined entity type set $E^{\mathcal{F}}$. Additionally, we possess a coarse-grained dataset $\mathcal{D}^{\mathcal{C}} = \{(\mathbf{X}_1^{\mathcal{C}}, \mathbf{Y}_1^{\mathcal{C}}), ..., (\mathbf{X}_{|\mathcal{D}^{\mathcal{C}}|}^{\mathcal{C}}, \mathbf{X}_{|\mathcal{D}^{\mathcal{C}}|}^{\mathcal{C}})\}$ characterized by a coarse-grained entity set $E^{\mathcal{C}}$ (i.e. $|E^{\mathcal{F}}| > |E^{\mathcal{C}}|$). A coarse-grained dataset typically

has a smaller number of types than a fine-grained dataset. Throughout this study, we use $\mathcal{F}$ and $\mathcal{C}$ to distinguish between these datasets. It should be noted that our method can be readily extended to accommodate multiple coarse-grained datasets, incorporating an intrinsic multi-level hierarchy. However, our primary discussion revolves around a single coarse-grained dataset for simplicity and readability.

## 3.2 Training CoFiNER

We aim to utilize both coarse- and fine-grained datasets directly in a single model training. Figure 1 illustrates an overview of the fine-grained NER model training process using both coarse- and fine-grained datasets. The CoFiNER training process consists of the following four steps:

**Step 1- Training a fine-grained model**. In the first step, we train a fine-grained model $f^{\mathcal{F}}(\theta)$ with the low-resource fine-grained dataset $\mathcal{D}^{\mathcal{F}}$. This process follows a typical supervised learning approach for NER. For a training example $(\mathbf{X}^{\mathcal{F}}, \mathbf{Y}^{\mathcal{F}})$, $\mathbf{X}^{\mathcal{F}}$ is fed into a PLM (Pre-trained Language Model), such as BERT, RoBERTa, to generate a contextual representations $\mathbf{h}_i \in \mathbb{R}^d$ of each token $x_i$.

$$\mathbf{H} = [\mathbf{h}_1, ..., \mathbf{h}_n] = \mathrm{PLM}([x_1^{\mathcal{F}}, ..., x_n^{\mathcal{F}}]). \quad (1)$$

Then, we apply a softmax layer to obtain the label probability distribution:

$$\mathbf{p}_i^{\mathcal{F}} = softmax(\mathbf{W}\mathbf{h}_i + \mathbf{b})$$

where $\mathbf{W} \in \mathbb{R}^{|E^{\mathcal{F}}| \times d}$ and $\mathbf{b} \in \mathbb{R}^{|E^{\mathcal{F}}|}$ represent the weights and bias of the classification head, respectively. To train the model using a fine-grained dataset, we optimize the cross-entropy loss function:

$$L_{\mathcal{F}} = -\frac{1}{n} \sum_{i=1}^{n} \log \mathbf{p}_i^{\mathcal{F}}[y_i^{\mathcal{F}}] \quad (2)$$

where $y_i \in E^{\mathcal{F}}$ is the fine-grained label for the token $x_i$.

**Step 2 - Generating an F2C matrix**. To fully leverage the hierarchy between coarse- and fine-grained entity types, we avoid training separate NER models for each dataset. Instead, we utilize a single model that incorporates an F2C mapping matrix that transforms a fine-grained output into a corresponding coarse-grained output. The F2C mapping matrix assesses the conditional probability of a coarse-grained entity type $s \in E^{\mathcal{C}}$ given

a fine-grained label $\ell \in E^{\mathcal{F}}$ (i.e., $\mathbf{M}_{\ell,s} = p(y^{\mathcal{C}} = s | y^{\mathcal{F}} = \ell)$).

Given a fine-grained probability distribution $\mathbf{p}_i^{\mathcal{F}}$ computed using the proposed model, the marginal probability of a coarse-grained type $s$ can be computed as

$$\mathbf{p}_i^{\mathcal{C}}[s] = \sum_{\ell \in E^{\mathcal{F}}} p(y^{\mathcal{C}} = s | y^{\mathcal{F}} = \ell) \cdot \mathbf{p}_i^{\mathcal{F}}[\ell].$$

Thus, the coarse-grained output probabilities are simply computed as follows:

$$\mathbf{p}_i^{\mathcal{C}} = \mathbf{p}_i^{\mathcal{F}} \cdot \mathbf{M} \quad (3)$$

where $\mathbf{M} \in \mathbb{R}^{|E^{\mathcal{F}}| \times |E^{\mathcal{C}}|}$ is the F2C mapping matrix whose row-wise sum is 1. By introducing this F2C mapping matrix, we can train a single model using multiple datasets with different granularity levels.

Manual annotation is a straightforward approach that can be used when hierarchical information is unavailable. However, it is not only cost-intensive but also noisy and subjective, especially when there are multiple coarse-grained datasets or a large number of fine-grained entity types. We introduce an efficient method for automatically generating an F2C matrix in §3.3.

**Step 3 - Filtering inconsistent coarse labels**. Although a fine-grained entity type is usually a subtype of a coarse-grained type, there can be some misalignments between the coarse- and fine-grained entity types. For example, an entity "Microsoft" in a financial document can either be tagged as `Company` or `Stock` which are not hierarchical. This inconsistency can significantly degrade the model's performance.

To mitigate the effect of inconsistent labeling, we devise an inconsistency filtering method aimed at masking less relevant coarse labels. By automatically filtering out the inconsistent labels from the coarse-grained dataset, we investigate the coarse-grained labels using the fine-grained NER model trained in Step 1. For each token in the coarse-grained dataset, we predict the coarse-grained label using the fine-grained model and the mapping matrix as follows:

$$\tilde{y}_i^{\mathcal{C}} = \arg\max \mathbf{p}_i^{\mathcal{C}}. \quad (4)$$

If the predicted label is the same as the coarse-grained label (i.e., $y_i^{\mathcal{C}} = \tilde{y}_i^{\mathcal{C}}$), we assume that the coarse-grained label is consistent with the fine-grained one and can benefit the model. Otherwise,

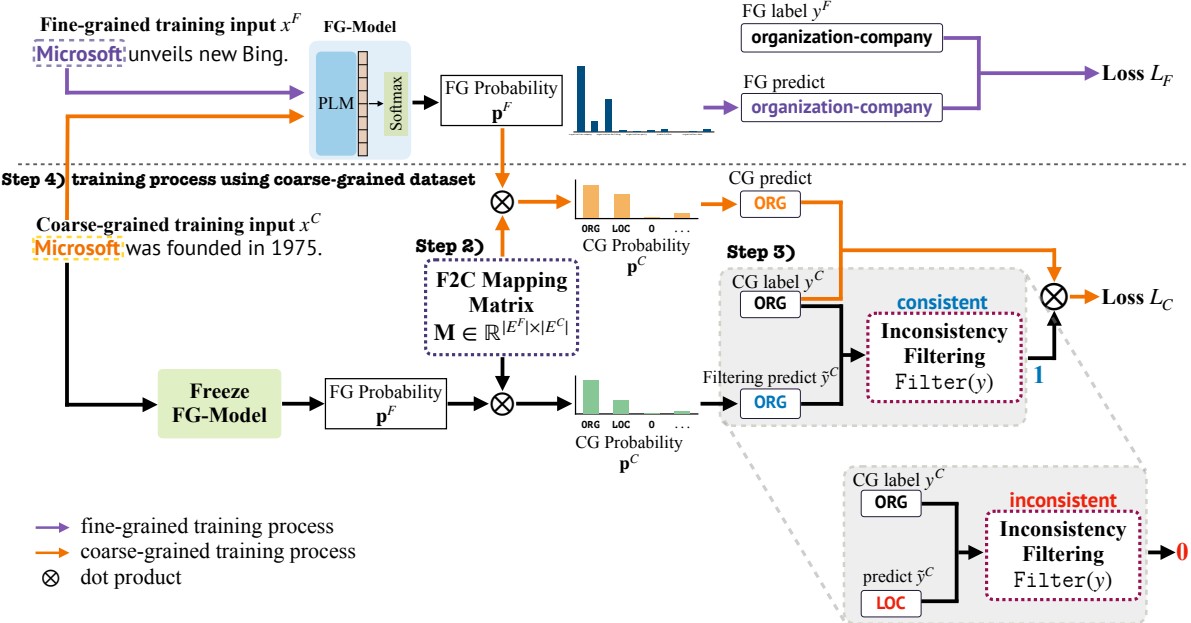

**Step 1) training process using fine-grained dataset**

Fine-grained training input $x^F$

Microsoft unveils new Bing.

FG-Model

PLM → Softmax → FG Probability $\mathbf{p}^F$

FG label $y^F$
organization-company

FG predict
organization-company

→ Loss $L_F$

**Step 4) training process using coarse-grained dataset**

Coarse-grained training input $x^C$

Microsoft was founded in 1975.

**Step 2)**
**F2C Mapping Matrix**
$\mathbf{M} \in \mathbb{R}^{|E^F| \times |E^C|}$

CG Probability $\mathbf{p}^C$  ORG LOC O ...

CG predict
ORG

**Step 3)**

CG label $y^C$
ORG

Filtering predict $\tilde{y}^C$
ORG

**consistent**
Inconsistency Filtering
Filter($y$)

1

→ Loss $L_C$

Freeze FG-Model → FG Probability $\mathbf{p}^F$ → CG Probability $\mathbf{p}^C$  ORG LOC O ...

CG label $y^C$
ORG

predict $\tilde{y}^C$
LOC

**inconsistent**
Inconsistency Filtering
Filter($y$)

→ 0

→ fine-grained training process
→ coarse-grained training process
⊗ dot product

Figure 1: Overall process of our proposed method.

we regard the label as inconsistent with fine-grained types and do not utilize the coarse-grained label in Step 4. Note that the fine-grained NER model is frozen during this phase.

**Step 4 - Jointly training CoFiNER with both datasets** . The model is trained by alternating between epochs using coarse- and fine-grained data. This is an effective learning strategy for training models using heterogeneous datasets (Jung and Shim, 2020).

For the fine-grained batches, CoFiNER is trained by minimizing the loss function defined in Equation (2), as described in Step 1. Meanwhile, we use the F2C mapping matrix to generate coarse-grained outputs and utilize inconsistency filtering for coarse-grained batches. Thus, we compute the cross-entropy loss between the coarse-grained label $y_i^C$ and the predicted probabilities $\mathbf{p}_i^C$ when the coarse-grained label is consistent with our model (i.e., $y_i^C = \tilde{y}_i^C$) as follows:

$$L_C = -\frac{1}{m} \sum_{i=1}^{m} \log \mathbf{p}_i^C[y_i^C] \cdot \mathbb{I}[y_i^C = \tilde{y}_i^C] \quad (5)$$

where $m$ is a length of $\mathbf{X}^C$. For example, suppose that the coarse label of token $x_i$ is $y_i^C =$ORG. If the estimated coarse label $\tilde{y}_i^C$ is ORG, the loss for the token is $-\log \mathbf{p}_i^C[y_i^C]$. Otherwise, it is zero.

### 3.3 Construction of the F2C mapping matrix

The F2C mapping matrix assesses the conditional probability of a coarse-grained entity type $s \in E^C$ given a fine-grained label $\ell \in E^F$ (i.e., $\mathbf{M}_{\ell,s} = p(y^C = s | y^F = \ell)$). As there are only a few identical texts with both coarse- and fine-grained annotations simultaneously, we can not directly calculate this conditional probability using the data alone. Thus, we approximate the probability using a coarse-grained NER model and fine-grained labeled data as follows:

$$\mathbf{M}_{\ell,s} = p(y^C = s | y^F = \ell) \approx p(\tilde{y}^C = s | y^F = \ell).$$

To generate the mapping matrix, we first train a coarse-grained NER model $f^C(\theta)$ with the coarse-grained dataset $\mathcal{D}^C$. Then, we reannotate the fine-grained dataset $\mathcal{D}^F$ by using the coarse-grained model $f^C(\theta)$. As a result, we obtain parallel annotations for both coarse- and fine-grained types in the fine-grained data $\mathcal{D}^F$. By using the parallel annotations, we can compute the co-occurrence matrix $\mathbf{C} \in \mathbb{N}^{|E^F| \times |E^C|}$ where each cell $\mathbf{C}_{\ell,s}$ is the number of tokens that are labeled as fine-grained type $\ell$ and coarse-grained type $s$ together.

Because some labels generated by the coarse-grained model $f^C(\theta)$ can be inaccurate, we refine the co-occurrence matrix by retaining only the top-$k$ counts for each fine-grained type and setting the rest to zero. Our experiments show that our model performs best when $k = 1$. This process effec-

tively retains only the most frequent coarse-grained categories for each fine-grained type, thereby improving the precision of the resulting mapping. Finally, we compute the conditional probabilities for all $\ell \in E^{\mathcal{F}}, s \in E^{\mathcal{C}}$ by using the co-occurrence counts as follows:

$$\mathbf{M}_{\ell,s} = p(\tilde{y}^{\mathcal{C}} = s | y^{\mathcal{F}} = \ell) = \frac{\mathbf{C}_{\ell,s}}{\sum_{s' \in E^{\mathcal{C}}} \mathbf{C}_{\ell,s'}}. \quad (6)$$

The F2C mapping matrix $\mathbf{M}$ is used to predict the coarse-grained labels using Equation (3).

## 4 Experiments

### 4.1 Datasets

| Dataset | # Sentences | | | # Types |
|---|---|---|---|---|
| | Train | Dev | Test | |
| Coarse-grained datasets | | | | |
| CoNLL'03 | 14k | 3.3k | 3.4k | 4 |
| OntoNotes | 59.9k | 8.5k | 8.3k | 18 |
| Fine-grained dataset Few-NERD | | | | |
| 10-Shot | 0.3k | | | |
| 20-Shot | 0.6k | | | |
| 40-Shot | 1.1k | 18.8k | 37.6k | 66 |
| 80-Shot | 2.2k | | | |
| 100-Shot | 2.7k | | | |

Table 1: Statistics of each dataset.

We conduct experiments using a fine-grained NER dataset, Few-NERD (SUP) (Ding et al., 2021), as well as two coarse-grained datasets, namely, OntoNotes (Weischedel et al., 2013) and CoNLL'03 (Tjong Kim Sang, 2002). The fine-grained dataset Few-NERD comprises 66 entity types, whereas the coarse-grained datasets CoNLL'03 and OntoNotes consist of 4 and 18 entity types, respectively. The statistics for the datasets are listed in Table 1.

$K$**-shot sampling for the fine-grained dataset.** Because we assumed a small number of examples for each label in the fine-grained dataset, we evaluated the performance in the $K$-shot learning setting. Although Few-NERD provides few-shot samples, they are obtained based on an $N$-way $K$-shot scenario, where $N$ is considerably smaller than the total number of entity types. However, our goal is to identify named entities across all possible entity types. For this setting, we resampled $K$-shot examples to accommodate all-way $K$-shot scenarios.

Since multiple entities exist in a single sentence, we cannot strictly generate exact $K$-shot samples for all the entity types. Therefore, we adopt the

$K \sim (K+5)$-shot setting. In the $K \sim (K+5)$-shot setting, there are at least $K$ examples and at most $K+5$ examples for each entity type. See Appendix A for more details. In our experiments, we sampled fine-grained training data for $K = 10, 20, 40, 80,$ and $100$.

### 4.2 Experimental Settings

In experiments, we use transformer-based PLM, including BERT$_{\mathsf{BASE}}$, RoBERTa$_{\mathsf{BASE}}$, and RoBERTa$_{\mathsf{LARGE}}$. In CoFiNER, we follow RoBERTa$_{\mathsf{LARGE}}$ to build a baseline model. The maximum sequence length is set to 256 tokens. The AdamW optimizer (Loshchilov and Hutter, 2019) is used to train the model with a learning rate of $2e{-}5$ and a batch size of 16. The number of epochs is varied for each model. We train the fine-grained model, CoFiNER, over 30 epochs. To construct the F2C mapping matrix, the coarse-grained model is trained for 50 epochs using both CoNLL'03 and OntoNotes. To train the inconsistency filtering model, we set different epochs based on the number of shots: For 10, 20, 40, 80, and 100 shot settings, the epochs are 150, 150, 120, 50, and 30, respectively. We report the results using span-level F1. The dropout with a probability of 0.1 is applied. All the models were trained on NVIDIA RTX 3090 GPUs.

### 4.3 Compared Methods

In this study, we compare the performance of CoFiNER with that of both the supervised and few-shot methods. We modified the existing methods for our experimental setup and re-implemented them accordingly.

**Supervised method.** We use BERT$_{\mathsf{BASE}}$, RoBERTa$_{\mathsf{BASE}}$, and RoBERTa$_{\mathsf{LARGE}}$ as the supervised baselines, each including a fine-grained classifier on the head. In addition, PIQN (Shen et al., 2022) and PL-Marker (Ye et al., 2022) are methods that have achieved state-of-the-art performance in a supervised setting using the full Few-NERD dataset. All models are trained using only a Few-NERD dataset.

**Few-shot method.** The LSFS (Ma et al., 2022a) leverages a label encoder to utilize the semantics of label names, thereby achieving state-of-the-art results in low-resource NER settings. The LSFS applies a pre-finetuning strategy to learn prior knowledge from the coarse-grained dataset, OntoNotes. For a fair comparison, we also conducted pre-

| Method | Model | 10-shot | 20-shot | 40-shot | 80-shot | 100-shot |
|--------|-------|---------|---------|---------|---------|----------|
| | BERT$_{BASE}$ (Devlin et al., 2019) | 23.101 | 34.718 | 43.138 | 46.182 | 46.449 |
| | RoBERTa$_{BASE}$ (Liu et al., 2019) | 21.073 | 34.157 | 39.226 | 45.735 | 46.942 |
| Supervised Method | RoBERTa$_{LARGE}$ (Liu et al., 2019) | 29.137 | 41.425 | 51.137 | 55.057 | 54.172 |
| | PIQN (Shen et al., 2022) | 21.750 | 22.007 | 28.533 | 29.339 | 38.658 |
| | PL-Marker (Ye et al., 2022) | 40.902 | 48.064 | 52.395 | 53.249 | 53.061 |
| Few-shot Method | LSFS (Ma et al., 2022a) | **47.998** | 43.269 | 50.595 | 51.420 | 50.366 |
| Proposed Method | CoFiNER | 44.951 | **51.142** | **56.409** | **56.847** | **57.178** |

Table 2: Results on few-shot NER. The best scores across all models are marked **bold**.

finetuning on OntoNotes and performed fine-tuning on each shot of the Few-NERD dataset.

**Proposed method**. We trained our CoFiNER model as proposed in §3. In each epoch, CoFiNER is first trained on two coarse-grained datasets: OntoNotes and CoNLL'03. Subsequently, it is trained on the fine-grained dataset Few-NERD. We used RoBERTa$_{LARGE}$ as the pre-trained language model for CoFiNER in Equation (1).

## 4.4 Main Results

Table 2 reports the performance of CoFiNER and existing methods. The result shows that CoFiNER outperforms both supervised learning and few-shot learning methods. Because supervised learning typically needs a large volume of training data, these models underperform in low-resource settings. This demonstrates that our method effectively exploits coarse-grained datasets to enhance the performance of the fine-grained NER model. In other words, the proposed F2C mapping matrix significantly reduces the amount of fine-grained dataset required to train supervised NER models. In particular, CoFiNER achieves significant performance improvements compared to the state-of-the-art model PL-Marker, which also utilizes the same pre-trained language model RoBERTa$_{LARGE}$ as CoFiNER.

The few-shot method LSFS yields the highest F1 score for the 10-shot case. However, this few-shot method suffers from early performance saturation, resulting in less than a 2.4 F1 improvement with an additional 90-shot. By contrast, the F1 score of CoFiNER increases by 12.2. Consequently, CoFiNER outperforms all the compared methods except for the 10-shot case. In summary, the proposed method yields promising results for a wide range of data sample sizes by explicitly leveraging the inherent hierarchical structure.

| Model | 10-shot | 20-shot | 40-shot | 80-shot | 100-shot |
|-------|---------|---------|---------|---------|----------|
| CoFiNER | **44.95** | **51.14** | 56.41 | 56.85 | **57.18** |
| w/o filtering | 41.38 | 43.87 | 53.42 | 54.70 | 55.13 |
| w/o OntoNotes | 42.63 | 44.51 | 57.18 | **56.97** | 55.54 |
| w/o CoNLL'03 | 43.99 | 50.07 | **57.54** | 56.71 | 56.27 |
| w/o coarse | 29.14 | 41.43 | 51.14 | 55.06 | 54.17 |

Table 3: Performances of ablation study over different components.

## 4.5 Ablation Study

An ablation study is conducted to validate the effectiveness of each component of the proposed method. The results are presented in Table 3. First, we remove the inconsistency filtering (*w/o filtering*) and observe a significant decrease in the F1 score, ranging from 2.05 to 7.27. These results demonstrate the effectiveness of our filtering method, which excludes mislabeled entities. Second, we provide the results using a single coarse-grained dataset (*w/o OntoNotes* and *w/o CoNLL'03*). Even with a single coarse-grained dataset, our proposed method significantly outperforms *w/o coarse*, which is trained solely on the fine-grained dataset (i.e. RoBERTa$_{LARGE}$ in Table 2).

This indicates the effectiveness of using a well-aligned hierarchy through the F2C mapping matrix and inconsistency filtering. Although we achieve a significant improvement even with a single coarse-grained dataset, we achieve a more stable result with two coarse-grained datasets. This implies that our approach effectively utilizes multiple coarse-grained datasets, although the datasets contain different sets of entity types.

## 4.6 Analysis

In this section, we experimentally investigate how the F2C mapping matrix and inconsistency filtering improve the accuracy of the proposed model.

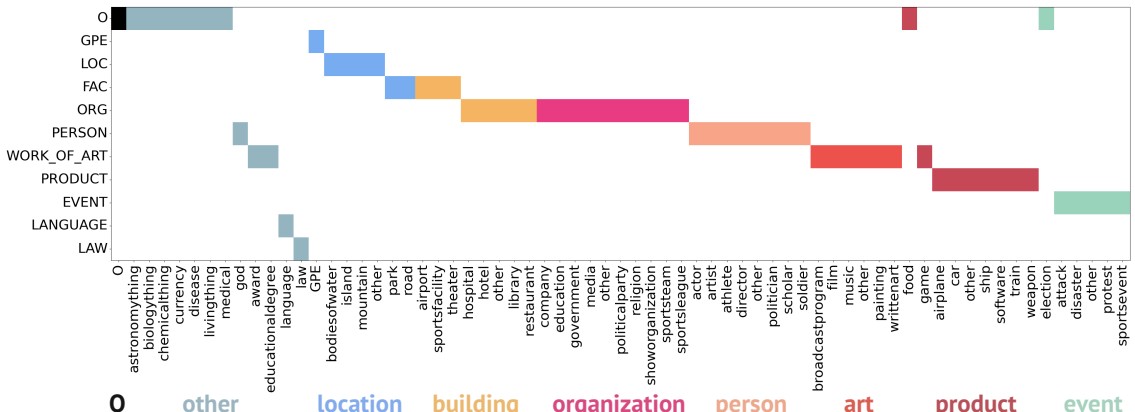

Figure 2: The F2C mapping matrix between OntoNotes and 100-shot of the Few-NERD dataset. During the generation of the F2C mapping matrix, some coarse-grained types are not mapped to any fine-grained types. The 7 unmapped types are not represented: DATE, TIME, PERCENT, MONEY, QUANTITY, ORDINAL, and CARDINAL.

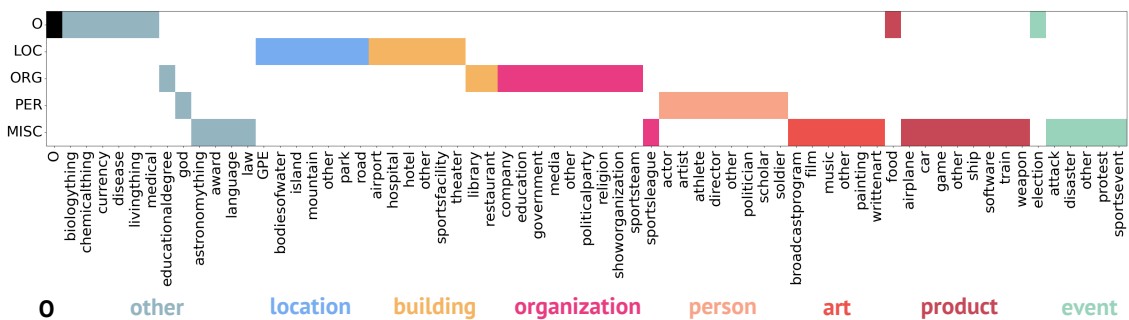

Figure 3: The F2C mapping matrix between CoNLL'03 and 100-shot of the Few-NERD dataset.

### 4.6.1 F2C Mapping Matrix

Mapping the entity types between the coarse- and fine-grained datasets directly affects the model performance. Hence, we investigate the mapping outcomes between the coarse- and fine-grained datasets. Figures 2 and 3 show the F2C matrices for FewNERD-OntoNotes and FewNERD-CoNLL'03, respectively. In both figures, the x- and y-axis represent the fine-grained and coarse-grained entity types, respectively. The colored text indicates the corresponding coarse-grained entity types in Few-NERD, which were not used to find the mapping matrix. The mapping is reliable if we compare the y-axis and the colored types (coarse-grained types in Few-NERD). Even without manual annotation of the relationship between coarse- and fine-grained entities, our method successfully obtains reliable mapping from fine-grained to coarse-grained types. Our model can be effectively trained with both types of datasets using accurate F2C mapping.

Figure 4 provides the F1 scores by varying the hyperparameter $k$ to refine the F2C mapping matrix described in §3.3. 'all' refers to the usage of the complete frequency distribution when creating an

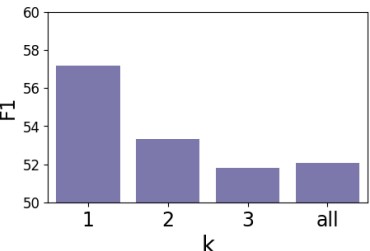

Figure 4: Impact of top-$k$ values in the F2C mapping matrix. F1 is reported on the test data.

F2C mapping matrix. We observe that the highest performance is achieved when $k$ is set to 1, and as $k$ increases, the performance gradually decreases. Although the optimal value of $k$ can vary depending on the quality of the coarse-grained data and the performance of the coarse-grained NER model, the results indicate that a good F2C mapping matrix can be obtained by ignoring minor co-occurrences.

Additionally, We conducted an experiment by setting the F2C mapping matrix to be learnable and comparing it with our non-learnable F2C matrix. The non-learnable approach showed better

| # | Type | Example |
|---|------|---------|
| **1** | predict | ... they may be offshoots of the **intifadah**EVENT, the **Palestinian rebellion**EVENT in ... |
|   | target | ... they may be offshoots of the intifadah, the **Palestinian**NORP rebellion in ... |
| **2** | predict | ... players heading to **Canada**GPE, particularly **Toronto**GPE for the **All-star**EVENT ... |
|   | target | ... players heading to **Canada**GPE, particularly **Toronto**GPE for the All-star ... |
| **3** | predict | Judges at **Flight 103 Lockerbie trial**EVENT are expected ... |
|   | target | Judges at Flight 103 **Lockerbie**GPE trial are expected ... |

Table 4: Inconsistent examples: The labeled sentence in the original coarse-grained dataset is the "target" type, while the label predicted by the model is the "predict" type. Consistent entity types are indicated in blue, while inconsistent entity types are indicated in red.

performance, hence we adopted this approach for CoFiNER. Detailed analysis and experiment results are in Appendix B.2.

### 4.6.2 Inconsistency Filtering

We aim to examine whether inconsistency filtering successfully screened out inconsistent entities between OntoNotes and Few-NERD datasets. To conduct this analysis, we describe three inconsistent examples. Table 4 shows the predicted values and target labels, which correspond to the coarse-grained output of the fine-grained NER model trained as described in §3.2 and the golden labels of the coarse-grained dataset.

The first example illustrates the inconsistency in entity types between mislabeled entities. In the original coarse-grained dataset, "Palestinian" is labeled as NORP, but the model trained on the fine-grained dataset predicts "Palestinian rebellion" as its appropriate label, EVENT. However, annotators of the OntoNotes labeled "Palestinian" as NORP, whereas the fine-grained NER model correctly predicts the highly informative actual label span. The inconsistency caused by a label mismatch between the coarse-grained and fine-grained datasets can result in performance degradation.

In the second example, both "Canada" and "Toronto" are consistently labeled as GPE; thus, the model is not confused when training on these two entities. However, in the case of "All-star", we can observe a mismatch. This example in the coarse-grained dataset is labeled O instead of the correct entity type EVENT, indicating a mismatch. Through inconsistency filtering, unlabeled "All-star" is masked out of the training process.

As shown in the examples, inconsistency filtering is necessary to mitigate the potential noise arising from mismatched entities. We analyze the filtering results for each coarse-grained label to assess its impact on model performance. Figure 5

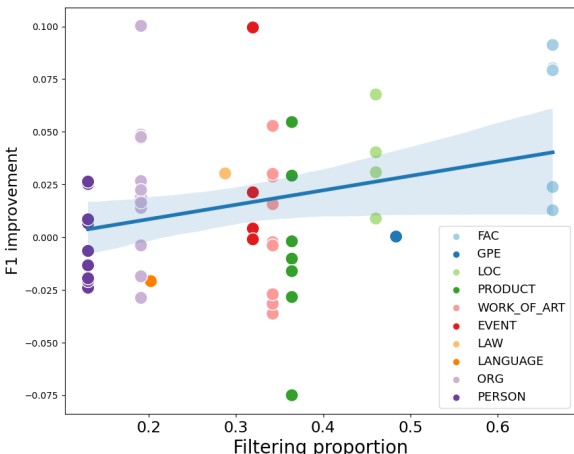

Figure 5: Performance with increasing filtering proportion between Few-NERD and OntoNotes datasets. The correlation coefficient between filtering proportion and performance improvement is 0.29.

illustrates the correlation between the filtering proportion and the performance improvement for each coarse-grained label mapped to the fine-grained labels. In this figure, a higher filtering proportion indicates a greater inconsistency between the two datasets. F1 improvements indicate the difference in performance when filtering is applied and when it is not. Each data point refers to a fine-grained label mapped onto a coarse-grained label. As the proportion of the filtered entities increases, the F1 scores also increase. These improvements indicate that inconsistency filtering effectively eliminates noisy entities, enabling the model to be well-trained on consistent data only.

## 5 Conclusion

We proposed CoFiNER, which explicitly leverages the hierarchical structure between coarse- and fine-grained types to alleviate the low-resource problem of fine-grained NER. We devised the F2C mapping matrix that allows for fine-grained NER model

training using additional coarse-grained datasets. However, because not all coarse-grained entities are beneficial for the fine-grained NER model, the proposed inconsistency filtering method is used to mask out noisy entities from being used in the model training process. We found through experiments that using a smaller amount of consistent data is better than using a larger amount of data without filtering, thus demonstrating the crucial role of inconsistency filtering. Empirical results confirmed the superiority of CoFiNER over both the supervised and few-shot methods.

## Limitations

Despite the promising empirical results of this study, there is still a limitation. The main drawback of CoFiNER is that the token-level F2C mapping matrix and inconsistency filtering may not be directly applicable to nested NER tasks. Nested NER involves identifying and classifying certain overlapping entities that exceed the token-level scope. Because CoFiNER addresses fine-grained NER at the token level, it may not accurately capture entities with nested structures. Therefore, applying our token-level approach to nested NER could pose challenges and may require further adaptations or other modeling techniques to effectively handle the hierarchical relations between nested entities.

## Acknowledgments

This work was supported by the National Research Foundation of Korea(NRF) grant funded by the Korea government(MSIT) (No. NRF-2022R1G1A1013549). This work was supported by Institute of Information & communications Technology Planning & Evaluation(IITP) grant funded by the Korea government(MSIT) (No. RS-2023-00261068, Development of Lightweight Multimodal Anti-Phishing Models and Split-Learning Techniques for Privacy-Preserving Anti-Phishing) and (No.RS-2022-00155885, Artificial Intelligence Convergence Innovation Human Resources Development (Hanyang University ERICA)). This research was also supported by the Ministry of Trade, Industry, and Energy (MOTIE), Korea, under the "Project for Research and Development with Middle Markets Enterprises and DNA(Data, Network, AI) Universities" (Smart Home Based LifeCare Service & Solution Development)(P0024555) supervised by the Korea Institute for Advancement of Technology (KIAT).

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

# Appendix

## A  Sampling Algorithm

---

**Algorithm 1** $K \sim (K+5)$ sampling algorithm

    **Input:** Dataset $\mathbf{X}$, labeled set $\mathbf{Y}$, $K$
    **Output:** Fine-grained dataset $D^{\mathcal{F}}$

---

1:   Sort classes in $\mathbf{Y}$ based on their freq. in $\mathbf{X}$
2:   $D^{\mathcal{F}} \leftarrow \emptyset$;        ▷ Init the train dataset
         ▷ Init the count of entity classes in $D^{\mathcal{F}}$
3:   **for** $i = 1$ to $|\mathbf{Y}|$ **do**
4:      Count $[i] = 0$;
5:   **for** $i = 1$ to $|\mathbf{Y}|$ **do**
6:      **if** $\forall$ Count$[i'] \geq K$ **then**
7:         break;
8:      Randomly sample $(x, y) \in \mathbf{X}$ s.t. $\mathbf{Y}_i \in y$;
9:      **for** $j = 1$ $to$ $|y|$ **do**
10:         Count$^{\mathbf{Y}}[j]$ += 1;
11:      **if** $\exists$ Count$[i]$ + Count$^{\mathbf{Y}}[i] > K + 5$ **then**
12:         Continue;
13:      **else**
14:         $D^{\mathcal{F}} \leftarrow D^{\mathcal{F}} \cup \{(x, y)\}$;
15:         Count $\leftarrow$ Count + Count$^{\mathbf{Y}}$;
16: **return** $D^{\mathcal{F}}$

---

Unlike other tasks, NER involves multiple entity occurrences within a sentence, making it too restrictive to sample an exact count. We adopt a

$K\sim(K+5)$-shot setting to minimize differences in the number of entity types. Additionally, we sample the entity types, starting from those with fewer occurrences to ensure a balanced distribution of multiple entity types within the sentences. Algorithm 1 presents the $K$-shot sampling algorithm used in this study.

## B  Additional Experiments

### B.1  Generalization of Inconsistency filtering in diverse dataset settings

| Setting | Coarse-grained datasets | |
|---|---|---|
| | Few-NERD_coarse | OntoNotes + CoNLL'03 |
| w/ filtering | 56.63 | 57.18 |
| w/o filtering | 56.46 | 55.13 |

Table 5: Results on different coarse-grained datasets. A fine-grained dataset is 100-shot of the Few-NERD.

To assess the generalization of the proposed method, we conducted experiments under various coarse- and fine-grained dataset settings.

First, we verified the robustness of inconsistency filtering through experiments using different coarse-grained datasets. The fine-grained dataset, Few-NERD, remains unchanged. Since Few-NERD has both coarse- and fine-grained labels, we constructed a coarse-grained dataset **Few-NERD_coarse** using the coarse-grained labels from Few-NERD. When compared to Few-NERD_coarse, entities in OntoNotes and CoNLL'03 are inconsistently labeled with Few-NERD because they were independently created. In Table 5, Few-NERD_coarse exhibits a higher F1 score in the *w/o filtering* setting due to its consistency with fine-grained labels of Few-NERD. However, the performance improvement achieved with filtering is more substantial in the inconsistent datasets, when compared to the consistent dataset. This result indicates that inconsistency filtering improves performance by filtering out the mismatching labels. Therefore, we have demonstrated the importance of using inconsistency filtering to filter out noise when working with datasets that employ different labeling schemes. Furthermore, by achieving effective performance improvements across various coarse-grained datasets, we have provided evidence of the robustness of the filtering method.

Second, we validate the generalization performance through experiments conducted in different coarse- and fine-grained dataset settings. We

| Model | F1 |
|---|---|
| RoBERTa$_{\text{LARGE}}$ | 75.15 |
| PL-Marker | 74.03 |
| CoFiNER | 80.44 |
| w/o filtering | 78.20 |
| w/o coarse | 75.15 |

Table 6: Performances of different models and ablation studies on our model. RoBERTa$_{\text{LARGE}}$ and *w/o coarse* are identical.

set up the CoNLL'03, which has 4 entity types, as the coarse-grained dataset, and the 100-shot OntoNotes, which has a finer label with 19 entity types, as the fine-grained dataset. In Table 6, when compared to the two top-performing models in the main results, RoBERTa$_{\text{LARGE}}$ and the state-of-the-art PL-Marker, CoFiNER show consistently higher performance. Furthermore, as shown in the ablation study that was conducted following the same methodology in §4.5, CoFiNER exhibited superior performance.

In conclusion, through above the two experiments, our method has been demonstrated to work robustly across different coarse- and fine-grained dataset settings.

### B.2  Comparison with learnable F2C mapping matrix

| Matrix Type | F1 |
|---|---|
| non-learnable | 56.27 |
| learnable | 53.25 |

Table 7: Results on learnable and non-learnable F2C mapping matrix on 100-shot of Few-NERD.

To find the optimal F2C mapping matrix, we conducted experiments to explore the impact of making the F2C mapping matrix learnable. We use Few-NERD as a fine-grained dataset and OntoNotes as a coarse-grained dataset. Table 7 shows no performance gains when the F2C mapping matrix was set to be learnable. We found that the learnable matrix tends to form a pattern similar to what is shown in Figure 4 with $k$=all. This result suggests that taking minor co-occurrences into account leads to an overall decrease in performance. Based on this analysis, the non-learnable mapping matrix is used in our experiments.