# OpenReview forum: "Enhancing Low-resource Fine-grained Named Entity Recognition by Leveraging Coarse-grained Datasets"
_EMNLP/2023/Conference — EMNLP 2023 Main_

### Official Review · Reviewer_Y7sL · 2023-08-03

**Soundness:** 4

**Excitement:**

4: Strong: This paper deepens the understanding of some phenomenon or lowers the barriers to an existing research direction.

**Paper Topic And Main Contributions:**

This paper studies the task of few-shot fine-grained entity typing. It proposes to utilize widely available coarse-grained datasets as additional knowledge to help the fine-grained task. By explicitly modeling the coarse-to-fine relations, the proposed methods achieve strong performance to baseline few-shot methods and another method also takes external coarse-grained data. Extensive experiment analysis shows the effectiveness of each proposed part and provides insights to the proposed model.

**Questions For The Authors:**

- I wonder whether the F2C matrix is learnable or not during the final training stage. Can you justify why it is or is not?
- Table 3 shows that more data does not always improve the final results and the behavior varies for different number of shots. Is there any good explanation on this?

**Reasons To Accept:**

- This paper proposes a way to utilize a coarse-grained labeled data to help fine-grained typing by explicitly modeling the coarse-to-fine relations which are then used to help integrate training on both coarse- and fine-grained data.
- There are insights into the coarse-to-fine correlations and the inconsistency filtering parts to demonstrate the motivations. The illustration of F2C mapping is intriguing.
- The proposed method achieves strong performance compared with baselines.

**Reasons To Reject:**

- It would be interesting to see results on more fine-grained datasets.

**Reproducibility:**

4: Could mostly reproduce the results, but there may be some variation because of sample variance or minor variations in their interpretation of the protocol or method.

**Reviewer Confidence:**

4: Quite sure. I tried to check the important points carefully. It's unlikely, though conceivable, that I missed something that should affect my ratings.

**Typos Grammar Style And Presentation Improvements:**

line 43: 'cost' -> 'costly'
line 88: 'properly' -> ?
line 239: 'a number' -> 'a large number'?

---

> ### Author Rebuttal · Authors · 2023-08-29
>
> # Response to Reviewer Y7sL
>
> We sincerely appreciate the time and effort that the reviewer dedicated to reading our paper and providing insightful feedback. Please find below our responses to the points raised in the review.
>
> ---
>
> **It would be interesting to see results on more fine-grained datasets.**
>
> **Response:** Thank you for your thoughtful suggestion to evaluate our model on different fine-grained datasets. We agree that doing so would strengthen the paper's contributions.
>
> To demonstrate the robustness of our model across diverse settings, we conducted additional experiments using the following datasets:
>
> - Fine-grained dataset: OntoNotes (19 entity types)
> - Coarse-grained dataset: CoNLL (4 entity types)
>
> We used the same sampling algorithm as presented in our paper to sample 100-shot examples from OntoNotes and conducted an ablation study. Here are the results for different models:
>
> | Model | 100-shot |
> | --- | --- |
> | CoFiNER | 80.44 |
> |   w/o filtering | 78.20 |
> |   w/o coarse | 75.15 |
>
> CoFiNER consistently improves performance across different fine-grained NER tasks, and we believe these results support that CoFiNER is generally applicable to various fine-grained datasets.
>
> ---
>
> **I wonder whether the F2C matrix is learnable or not during the final training stage. Can you justify why it is or is not?**
>
> **Response:** Thank you for your question about the learnability of the F2C mapping matrix during the final training stage.
>
> In response to your question, we conducted supplementary experiments to explore the impact of making the F2C mapping matrix learnable. Unfortunately, our subsequent testing showed no performance gains when the F2C mapping matrix was set to be learnable. We conducted experiments under the following settings:
>
> - Fine-grained dataset: Few-NERD
> - Coarse-grained dataset: OntoNotes
>
> | F2C Mapping Matrix | 100-Shot |
> | --- | --- |
> | non-learnable | 56.27 |
> | learnable | 53.25 |
>
> We found that when the matrix is learnable, it tends to form a pattern similar to what is shown in Figure 4 with $K$=all. These results suggest that taking minor co-occurrences into account leads to an overall decrease in performance.
>
> Based on this additional analysis, we would like to maintain our original approach of using a fixed F2C mapping matrix. Even without the training through backpropagation, the F2C mapping matrix can capture the adequate relation between types, and in doing so, it contributes to a more stable and improved performance.
>
> Your question gave us the opportunity for further investigation, and we are grateful for that.
>
> ---
>
> **Table 3 shows that more data does not always improve the final results and the behavior varies for different number of shots. Is there any good explanation on this?**
>
> **Response:** As you pointed out, some models do not show a consistent improvement in performance with an increase in the number of shots. We believe the following are possible reasons for the behavior:
>
> **1. Overfitting**
>
> In few-shot settings, the risk of overfitting is inherently higher due to the limited amount of data. This is especially apparent in our experiments that exclude the OntoNotes or CoNLL datasets, making the models more prone to overfitting. This could possibly explain the fluctuating performance as the number of shots changes.
>
> On the other hand, CoFiNER leverages a more diverse set of training datasets, which helps to alleviate the risk of overfitting to some degree. Thus, it shows a consistent performance improvement as the number of shots increases.
>
> **2. A Mismatch of train and test data distributions**
>
> With a low number of shots, the limited training samples might be insufficient to capture the full range of the test data distribution.
>
> In the test dataset, the frequency of each entity type varies considerably. For instance, in the Few-NERD test set,  the fine-grained type `location-GPE` appears 20,409 times, while the type `art-painting` shows up just 58 times. However, when generating training data in few-shot setting, we try to make the occurrence of each entity type as uniform as possible. This can lead to a mismatch between the train and the test set. Consequently, as the amount of training data increases, there might be fluctuations in performance improvements on the test set.
>
> We hope that we have addressed your concerns regarding the observed behavior and provided a reasonable explanation.
>
> ---
>
> **line 43: 'cost' -> 'costly' line 88: 'properly' -> ? line 239: 'a number' -> 'a large number'?**
>
> **Response:** Thank you for the detailed comments. We will rectify them in our revision. Furthermore, we thoroughly proofread the manuscript and corrected grammatical errors and ambiguous expressions to improve readability.

---

### Official Review · Reviewer_ZwPd · 2023-08-04

**Soundness:** 3

**Excitement:**

3: Ambivalent: It has merits (e.g., it reports state-of-the-art results, the idea is nice), but there are key weaknesses (e.g., it describes incremental work), and it can significantly benefit from another round of revision. However, I won't object to accepting it if my co-reviewers champion it.

**Paper Topic And Main Contributions:**

Paper Topic: Fine-grained NER

Main Contribution (NLP engineering experiment): The authors propose to utilize the hierarchical structure between coarse-grained and fine-grained types to alleviate the low-resource problem in fine-grained NER. Experiments demonstrate the effectiveness of the proposed approach.

**Reasons To Accept:**

- Regarding the writing: The paper is well-written and easy to follow.
- Regarding the methodology: The method is well-designed to improve the low-resource problem in fine-grained NER, where a mapping matrix is constructed to measure the relations between the coarse- and fine-grained NE types and a filtering method is used to filter those inconsistent labels.
- Regarding the experiments: The experiments are sufficient and show the effectiveness of the proposed approach.

**Reasons To Reject:**

- The **hierarchical structure** is not well-described in the paper. It seems it is only the direct mapping between coarse- and fine-grained types instead of hierarchy.

**Reproducibility:**

4: Could mostly reproduce the results, but there may be some variation because of sample variance or minor variations in their interpretation of the protocol or method.

**Reviewer Confidence:**

3: Pretty sure, but there's a chance I missed something. Although I have a good feel for this area in general, I did not carefully check the paper's details, e.g., the math, experimental design, or novelty.

---

> ### Author Rebuttal · Authors · 2023-08-29
>
> # Response to Reviewer ZwPd
>
> We sincerely appreciate the time and effort that the reviewer dedicated to reading our paper and providing insightful feedback. Please find below our response to the points raised in the review.
>
> ---
>
> **The hierarchical structure is not well-described in the paper. It seems it is only the direct mapping between coarse- and fine-grained types instead of hierarchy.**
>
> **Response:**  We agree with your assessment that the paper may give the impression of direct mapping between coarse- and fine-grained types because we mainly introduce our model with a pair of coarse- and fine-grained datasets for readability. Furthermore, fully exploiting the hierarchical structure is important to achieve a high accuracy in fine-grained NER.
>
> In fact, our approach can straightforwardly leverage a multi-level hierarchy by incorporating multiple coarse-grained datasets during the training process as discussed in Section 3.1. In addition, we demonstrated performance improvements by using a multi-level hierarchy that incorporates both OntoNotes and CoNLL datasets, outperforming the results when employing a two-level hierarchy using either OntoNotes or CoNLL.
>
> We acknowledge that this could have been more clearly described in our paper, and we will clarify it in the revised manuscript.
>
> We would like to explain the multi-level hierarchy with `LOC` as an example. `LOC` type in the most coarsely grained CoNLL dataset, exhibits hierarchical relations with `GPE`, `LOC`, and `FAC` in the finer dataset, OntoNotes. These types are further divided into more specific types and establish correlations with the most finely grained dataset, Few-NERD. As illustrated in Figure 2,  `GPE`, `LOC`, and `FAC` in OntoNotes show hierarchical relations with 10 Few-NERD types, including `bodiesofwater`, `island`, `mountain`, `park`, and `road`.
>
> Despite the datasets exhibiting multi-level hierarchical relationships, we do not explicitly consider the relationships between the coarse-grained datasets, since our primary goal is to improve fine-grained NER.

---

### Official Review · Reviewer_D3qZ · 2023-08-10

**Soundness:** 3

**Excitement:**

4: Strong: This paper deepens the understanding of some phenomenon or lowers the barriers to an existing research direction.

**Paper Topic And Main Contributions:**

The paper details a method to use existing coarse-grained data to improve the probability of few-shot fine grained system. The system consists of a pipeline where fine-grained fine tuned model is used to produce the labels, which are then mapped to a coarse-grained label, these labels are filtered for inconsistency and loss is computed over only the consistent labels.
They use CoNLL and OntoNotes as the source coarse grained datasets to improve performance over few NERD. Results show they achieve improvements over baselines.

**Questions For The Authors:**

1) The method only back propagates loss for the examples for which the coarse label matches the truth. Why do you think this is helpul in training the model? Seems like the model does not get to learn from its mistakes.
2) I am not sure if I fully understand the w/o course baseline. is it not a supervised fine-tuned model then?
3) It could not sing this detail in the paper - which setting of Few NERD is used - SUP, INTER or INTRA?
4) Did the authors re-implement/ retrain all baselines with new splits? The PIQN numbers are much lower, would you know why?
5) OntoNotes has 18 types, the paper mentions 11. Which types are not used. Pls include details.

**Reasons To Accept:**

The task is interesting and useful - to leverage existing data for few-shot fine-grained
The paper is generally well written, with easy to understand figures and table. Improvements could be made as suggested.
The results show the method works. If the authors make the code available, this could be a useful repo.

**Reasons To Reject:**

Pls see the questions section - I have questions over the evaluation and also on why the method works when the loss erroneous labels is not back-propagated.

**Reproducibility:**

3: Could reproduce the results with some difficulty. The settings of parameters are underspecified or subjectively determined; the training/evaluation data are not widely available.

**Reviewer Confidence:**

4: Quite sure. I tried to check the important points carefully. It's unlikely, though conceivable, that I missed something that should affect my ratings.

---

> ### Author Rebuttal · Authors · 2023-08-29
>
> # Response to Reviewer D3qZ
>
> We sincerely appreciate the time and effort that the reviewer dedicated to reading our paper and providing insightful feedback. Please find below our responses to the points raised in the review.
>
> ---
>
> **The method only back propagates loss for the examples for which the coarse label matches the truth. Why do you think this is helpul in training the model? Seems like the model does not get to learn from its mistakes.**
>
> **Response:**  We would like to point out a key advantage of the inconsistency filtering through selective backpropagation: it mitigates potential confusion that could arise from label inconsistencies across datasets. If the inconsistency filtering is not employed, the model could learn wrong correlations from misaligned labels, potentially diminishing the benefits of leveraging multiple datasets.
>
> To empirically assess the impact of label inconsistencies on model performance, we conducted additional experiments using coarse-grained labels of Few-NERD as an auxiliary coarse-grained dataset.
>
> |  | Few-NERD_coarse | OntoNotes + CoNLL |
> | --- | --- | --- |
> | w/ Filtering | 56.63 | 57.18 |
> | w/o Filtering | 56.46 | 55.13 |
>
> Since Few-NERD dataset has both fine-grained and coarse-grained labels for every input text, the coarse-grained labels are consistent with the fine-grained labels. Meanwhile, OntoNotes and CoNLL, which were created independently with Few-NERD, have many inconsistent labels. The performance improvement due to the inconsistency filtering is greater in the inconsistent datasets, CoNLL and OntoNotes than in the consistent dataset, Few-NERD_coarse.
>
> The result implies that the inconsistency filtering improves performance by filtering out the inconsistent labels via selective backpropagation. Hence, we believe that erroneous labels could lead to confusion, as they deviate from the labeling scheme of Few-NERD, rather than helping the model learn from its mistakes.
>
> ---
>
> **I am not sure if I fully understand the w/o course baseline. is it not a supervised fine-tuned model then?**
>
> **Response:**  Yes, it is a supervised fine-tuned model. Thank you for pointing out the ambiguously described part of our paper. To clarify, *w/o coarse* baseline is a supervised fine-tuned model that is trained only on the fine-grained dataset, without incorporating any data from the coarse-grained datasets. Thus, the *w/o coarse* baseline is identical to RoBERTa_LARGE in Table 2 since we use RoBERTa_LARGE as the base model.
>
> We will make clear this part in the revised version.
>
> ---
>
> **It could not sing this detail in the paper - which setting of Few NERD is used - SUP, INTER or INTRA?**
>
> **Response:**  We used the dataset of the Few-NERD SUP setting. The INTER and INTRA settings described in the Few-NERD[1] use an $N$-way $K$-shot setting where the types in the train, dev, and test sets are all different.
>
> In contrast, we conducted our experiments in an **a*ll*-way $K$-shot setting**. We believe this approach is more **realistic** and **challenging** compared to $N$-way $K$-shot setting. $N$ usually represents a small, restricted number of classes, making it inadequate to handle a large number of entities. In our approach, we expand $N$ to include a much larger set of fine-grained labels, such as the 67 types in Few-NERD. This **a*ll*-way $K$-shot setting** makes our experimental setting more representative of real-world scenarios.
>
> ---
>
> **Did the authors re-implement/ retrain all baselines with new splits?**
>
> **Response:**  Yes, we re-implemented and retrained all baselines using new splits.
>
> We used the Hugging Face library to obtain pre-trained models for BERT and RoBERTa. For other baselines(PIQN, PL-Marker, and LSFS), we adapted the original code available on each paper's GitHub repositories, modified it for our experimental setup, and retrained them accordingly.
>
> **The PIQN numbers are much lower, would you know why?**
>
> **Response:** PIQN was originally proposed for a standard supervised setting, as demonstrated in their paper[2]. In this context, PIQN is trained on all available training data in the Few-NERD SUP setting. On the other hand, our work operates in an *all*-way $K$-shot setting, where we have limited examples for each class. PIQN, which is designed to perform well with a substantial amount of data, exhibits lower performance in our few-shot setting.
>
> ---
>
> **OntoNotes has 18 types, the paper mentions 11. Which types are not used. Pls include details.**
>
> **Response:**  As you mentioned, Figure 2 illustrates 11 out of 18 entity types.
>
> For clarity, we excluded certain coarse-grained types that were **not mapped** to any fine-grained types during the generation of the F2C mapping matrix using OntoNotes and Few-NERD datasets. The 7 unmapped types from OntoNotes are as follows: `DATE`, `TIME`, `PERCENT`, `MONEY`, `QUANTITY`, `ORDINAL`, and  `CARDINAL`. Because the correlation between coarse- and fine-grained types makes mapping decisions, consequently affects the types within the F2C mapping matrix.
>
> We hope this clarifies any confusion. We will also include these details in the revised manuscript.
>
> ---
>
> **References**
>
> [1] [Few-NERD: A Few-shot Named Entity Recognition Dataset](https://aclanthology.org/2021.acl-long.248) (Ding et al., ACL-IJCNLP 2021)
>
> [2] [Parallel Instance Query Network for Named Entity Recognition](https://aclanthology.org/2022.acl-long.67) (Shen et al., ACL 2022)

---

### Meta-Review · Area_Chair_314G · 2023-09-20

**Recommendation:** 4

**Metareview:**

This paper utilizes coarse-grained datasets for fine-grained NER with a small number of fine-grained annotations. The research questions have significant value and this paper gives the effective method. I suggest the authors give detailed descriptions for the method, and make precise revision accroding to comments of reviewers and responses.

---

### Decision · Program_Chairs · 2023-10-07

**Decision:**

Accept-Main

**Comment:**

This paper utilizes coarse-grained datasets for fine-grained NER with a small number of fine-grained annotations. The research questions have significant value and this paper gives the effective method. I suggest the authors give detailed descriptions for the method, and make precise revision accroding to comments of reviewers and responses.